# Integrating nested data into knowledge graphs with RML fields

Thomas Delva[(✉)][0000−0001−9521−2185], Dylan Van Assche[0000−0002−7195−9935], Pieter Heyvaert[0000−0002−1583−5719], Ben De Meester[0000−0003−0248−0987], and Anastasia Dimou[(✉)][0000−0003−2138−7972]

IDLab, Department of Electronics and Information Systems, Ghent University - imec
Technologiepark-Zwijnaarde 122, 9052 Ghent, Belgium
{firstname.lastname}@ugent.be

**Abstract.** To support business decisions or improve operational efficiency, heterogeneous data is often integrated into a knowledge graph. This integration can be achieved with one of the existing declarative mapping languages, which offer declarative data integration in the form of knowledge graphs. However, current mapping languages cannot always integrate data with nested structure, such as JSON or XML files or JSON documents stored in a database column. We designed a backwards-compatible extension of the RDF Mapping Language (RML) which empowers it to integrate nested data: RML fields. In this paper, we introduce RML fields, compare it with the state of the art in mapping languages, and validate it on mapping challenges formulated by the Knowledge Graph Construction W3C community group. Our extension allows addressing several of the challenges related to nested data that were previously not possible. RML fields can integrate even more datasets into knowledge graphs with all the advantages of using a language specially designed for that purpose. Our extension intends integrating multiple data sets independently, but some use cases require joins or other operations during knowledge graph generation, which we will investigate in the future.

## 1 Introduction

Graph structures recently became a popular way [10] to organize information: the so-called knowledge graphs [11]. Declarative mapping languages are often used to integrate non graph data into a knowledge graph [5]. A declarative mapping language allows describing schema and data transformations. R2RML, the W3C-recommended declarative mapping language creates knowledge graphs from tabular input data in databases [7]. R2RML was quickly extended to cover more input formats [8], but it also comes with added challenges.

References to common data formats like JSON or XML may return multiple values and these values can be composite: they may again contain multiple values. In contrast, a reference to tabular data typically returns exactly one, non-composite value: the value in a table cell. These two things, multiple and/or composite values, can occur independently of each other: a reference could

return one value composed of different attributes, it could return multiple non-composite values, such as integers, or it could return any other combination of multiple and composite values. For instance, multiple objects are returned by applying the JSONPath reference `$.characters.[*]` on the JSON document in fig. 1a and each returned object is itself composed of several attributes: `firstname` and `items`. Declarative mapping languages that integrate such formats as JSON or XML use references that return multiple and composite values, but current mapping languages do not completely handle this. That is why several[1] of the challenges[2] the KGC W3C Community Group identified are related to handling references that can return multiple and/or composite values.

Integrating mixed-format data faces a similar challenge: what if data in one format contains multiple or composite values stored in another format? Examples are JSON objects stored inside a database column (fig. 1b) or multiple values stored as a delimiter-separated string. While certainly unnormalized (it violates the first normal form for relational databases [3]) such data is not unrealistic.

We extended the RDF Mapping Language (RML) [8], which already allows integration of heterogeneous data, with a nested iteration model. The nested iteration model empowers RML to write nested loops over input data. Nested iterations solve both previously mentioned problems: (i) references returning multiple or composite values can be treated as a deeper iteration level and (ii) every iteration level can iterate over data in a different format.

The rest of the paper is structured as follows. We give an overview of how current mapping languages integrate nested data in section 2. Then, we introduce the "fields" extension to RML and show how nested fields allow integrating nested data in section 3. We show how RML fields can handle the challenges related to integrating nested data formulated by the W3C Knowledge Graph Construction Community Group in section 4. Finally we conclude in section 5.

---

[1] access-fields-outside-iteration, generate-multiple-values, multivalue-references, process-multivalue-reference and rdf-collections

[2] `https://github.com/kg-construct/mapping-challenges/`

```
{ "characters": [{
    "firstname": "Ash",
    "items": [
      {"name":"gloves", "weight":340},
      {"name":"sword", "weight":4400}
    ]}, {
    "firstname": "Misty",
    "items":[
      {"name":"gloves", "weight":340},
      {"name":"mittens", "weight":300},
      {"name":"hat", "weight":800}
    ]} ]}
```

(a) Example of tree-structured data in the JSON format.

| firstname; | items |
|---|---|
| Ash; | [{"name": "gloves", "weight": 340 }, {"name": "sword", "weight": 4400 }] |
| Misty; | [{"name": "gloves", "weight": 340 }, {"name": "mittens", "weight": 300 }, {"name": "hat", "weight": 800}] |

(b) Example of mixed-format data: JSON object stored in a CSV column.

```
:people/Ash/items/gloves    :weight 340  .
:people/Ash/items/sword     :weight 4400 .
:people/Misty/items/gloves  :weight 340  .
:people/Misty/items/mittens :weight 300  .
:people/Misty/items/hat     :weight 800  .
```

(c) This graph cannot be created from fig. 1a or fig. 1b with current languages, as it mixes data from multiple hierarchical levels (**bolded**).

Fig. 1: Current mapping languages cannot successfully handle this nested data.

## 2 Related work

With the increasing prevalence of RDF as a format for data on the web, W3C sought to standardize the RDF generation procedure. To this end, two recommendations were published related to generating RDF from relational databases: the Direct Mapping [1] and R2RML [7] recommendations. Direct Mapping is a transformation that generates an RDF graph with the same structure and contains exactly the same information as a relational database. R2RML is a declarative mapping language that can be used to define customized mappings from a relational database to RDF. With R2RML, information in a database can be used to generate RDF graphs with different structures than the database itself.

R2RML was soon generalized by RML [8] aiming to be extensible towards other input data formats than relational databases. RML provides examples for common formats like CSV, JSON and XML. To achieve this, RML introduces, among other things, the concept of reference formulation. A reference formulation is specified for each integrated data set to specify how data elements in that data set should be referred to: for example, RML uses by default (i) XPath expressions to refer to data in XML format, (ii) column names to refer to data in CSV/TSV format or relational databases, and (iii) JSONPath expressions to refer to data in JSON format. However, in going beyond relational databases, references to non-relational data that return multiple values are not considered, even though they may be needed for XML and JSON. Other mapping languages, such as xR2RML [14] and ShExML [9], were proposed to cover some of RML's limitations but none offers a complete solution.

xR2RML [14] extends both R2RML and RML and was the first to handle challenges that come with nested input data. For this reason, xR2RML introduced the nested term map and mixed-syntax paths.

Nested term maps can be used to generate triples from hierarchical data, where one of the triples' terms is generated from a deeper level of the input data's hierarchy. However, it becomes difficult to refer to data stored in different hierarchical levels in the input data. Therefore, xR2RML introduced the `xrr:pushDown` term, which allows to "push down" values from a higher hierarchical level into a lower hierarchical level. For example, in fig. 2, a nested term map is used together with `xrr:pushDown` to generate URIs from data on different hierarchical levels: `:people/Ash/items/gloves` is created by pushing `Ash` from level one in fig. 4a down to level two (inside the nested `items` array), where it can be used together with `gloves` to generate the needed URI. `xrr:pushDown` can be used to solve many practical cases, but, as the nested term map by definition generates individual terms, it does not account for cases where data from different hierarchical levels is used to generate *more than one term* in a triple, as in the graph in fig. 1c. There, `Ash`, `gloves`, and `340` come from more than one hierarchical level and are used in the subject *and* in the object. Therefore it is impossible to generate this graph using xR2RML.

Independently of nested term maps, xR2RML introduced mixed-syntax paths. These paths can be used to refer to data stored in mixed formats. An

```
<#Characters>
  a rr:TriplesMap;
  xrr:logicalSource [
    xrr:query """db.characters.find()""" ;
    rml:iterator "$.characters[*]" ] ;
  rr:subjectMap [
    rr:template ":people/{$.firstname}/" ] ;
  rr:predicateObjectMap [
    rr:predicate ex:hasItem;
    rr:objectMap [
      xrr:reference "$.items[*]" ;
      xrr:pushDown [
        xrr:reference "$.firstname" ;
        xrr:as "firstname" ] ;
      xrr:nestedTermMap [
        rr:template
          ":people/{$.firstname}/items/{$.name}"
      ] ] ] .
```

```
:people/Ash    :hasItem :people/Ash/items/gloves ,
                        :people/Ash/items/sword .
:people/Misty :hasItem :people/Misty/items/gloves ,
                        :people/Misty/items/mittens ,
                        :people/Misty/items/hat .
```

Fig. 2: This xR2RML mapping (left) partially handles the example in fig. 1: xR2RML can generate *single terms* from data from across the input hierarchy (shown **bolded** on the right), but not full triples, as is needed in fig. 1c.

```
ITERATOR chars_it <jsonpath: $.characters[*]> {
  PUSHED_FIELD firstname <firstname>
  ITERATOR items <items[*]> {
    FIELD name <name>
    FIELD weight <weight>
    POPPED_FIELD firstname <firstname> }}

EXPRESSION chars <chars_file.chars_it>

:Item :[chars.items.name] {
  :hasweight [chars.items.weight] ;
  :ownedBy :[chars.items.firstname] }
```

```
:gloves  :hasWeight 340 ;
         :ownedBy   :Ash .
:sword   :hasWeight 4400 ;
         :ownedBy   :Ash .
:gloves  :hasWeight 340 ;
         :ownedBy   :Misty .
:mittens :hasWeight 300 ;
         :ownedBy   :Misty .
:hat     :hasWeight 800 ;
         :ownedBy   :Misty .
```

Fig. 3: This ShExML mapping (left) partially handles the example in fig. 1: ShExML can access all the attributes required to generate the triples in fig. 1c, but can only make terms from exactly one attribute (shown **bolded** on the right).

example explains the idea best: if JSON objects are stored in a database column, fields of such a JSON object can be referred to with an expression like `Column(.)/JSONPath(.)`.

ShExML [9] uses ShEx shapes [15] to define the structure of RDF generated from other sources. To extract information from input data, ShExML uses iterators and fields. Iterators give a name to collections in the input data, and fields give a name to individual values. Iterators can be defined nestedly to handle nested input data. Names of fields and iterators are used in ShEx shape templates to specify how the extracted information is written to RDF. For referring to data in different hierarchical levels ShExML introduces "pushed" and "popped" fields which can push down information during nested iteration, similar to xR2RML's `xrr:pushdown`. As such, ShExML is missing little to generate the graph in fig. 1c, yet ShExML can only generate URIs from one attribute, while the desired URI `:people/Ash/items/gloves` is generated from two attributes. In fig. 3 we give a partial solution in ShExML for the input data and desired graph in fig. 1. ShExML does also not provide solutions for input data in mixed formats.

| Task Language | Referring to mixed-format data | Referring to tree-structured data | Writing nested data to graph |
|---|---|---|---|
| **xR2RML** | Mixed syntax paths | Nested term map | |
| **ShExML** | – | Nested iterator | Linked shapes |
| **RML fields** | Reference formulation | Nested fields | (Nested) term map |

Table 1: Overview of how different mapping languages handle different tasks related to generating graphs from nested data.

In the next section, we will build on xR2RML's concepts of nested term map and mixed-syntax paths and on ShExML's concepts of fields and nested iterators. Our main contribution on top of these two mapping languages is a method to preserve the relation between related values from different hierarchical levels without explicitly pushing down those values. The relation between xR2RML, ShExML and our contribution, RML fields, is shown in table 1.

## 3   RML fields

In this section we introduce the fields extension of RML. We will first explain how to extract information from nested data using fields. Then we show an algorithmic representation of the extracted information and how that information can be written to RDF. We will close the section by showing precisely how RML with fields is compatible with regular RML.

### 3.1   Fields

A field gives a name to a reference, as in ShExML. References are the part of RML for extracting information from data. This extraction involves two other concepts, besides the reference concept: iterators and records. The extraction process as it is in RML can be summarized as follows. Given a data source, the iterator extracts a list of records from it. From each record in this list, a reference extracts a value to create RDF terms with. For example the iterator `$.characters[*]` returns a list containing the objects in the `characters` JSON array in fig. 4a. The reference `$.firstname` extracts the `firstname` attribute of each record: in consecutive iterations this reference will return `Ash` and `Misty`. An RML field can be used to give a human-readable name to the extracted attribute, the field declaration for this is shown on lines 5-7 of fig. 4a. This name can then be used in term maps and URI templates to generate RDF triples from the extracted information.

The extraction scheme as it is explained so far does not specify yet what happens with references which return multiple or composite values. For example the JSONPath expression `$.items.[*]` will return each value in the items array and each of these returned values is themselves an object composed of attributes name and weight. Therefore we need to make this clarification about references:

a reference does not extract a value from a record, but a reference extracts *a list of records* from a record. In particular we treat extracted non-composite values, such as the string `Ash` or number `340`, as one particularly simple type of record.

This generalization of the reference concept is crucial, as it allows to define references that extract information from the output of other references, as that output is also just records. As such, it is possible to extract the name and weight information from the output of the `$.items.[*]` reference by using the `$.name` and `$.weight references` on the output of `$.items.[*]`. We introduce nested fields for such "chaining" of references and an example of a field with two nested fields inside is shown on lines 8-17 of fig. 4a. The relation between data sources, iterators, records and references is pictured in fig. 5.

```
:source1 a rml:LogicalSource ;
  rml:source :file ;
  rml:referenceFormulation ql:JSONPath ;
  rml:iterator "$.characters[*]" ;
  rml:field [
    rml:name "name" ;
    rml:reference "$.firstname" ; ] ;
  rml:field [
    rml:name "item" ;
    rml:reference "$.items[*]" ;
    rml:field [
      rml:name "name" ;
      rml:reference "$.name" ; ] ;
    rml:field [
      rml:name "weight" ;
      rml:reference "$.weight" ; ] ;
  ].
```

(a) RML fields snippet to extract information from the nested JSON collections in fig. 1a.

```
:source1 a rml:LogicalSource ;
  rml:source :file ;
  rml:referenceFormulation ql:CSV ;
  rml:field [
    rml:name "name" ;
    rml:reference "firstname" ; ] ;
  rml:field [
    rml:name "items" ;
    rml:reference "items" ;
  rml:field [
    rml:referenceFormulation ql:JSONPath ;
    rml:name "item" ;
    rml:reference "$.[*]" ;
    rml:field [
      rml:name "name" ;
      rml:reference "$.name" ; ] ;
    rml:field [
      rml:name "weight" ;
      rml:reference "$.weight" ; ] ;
  ] .
```

(b) RML fields snippet to extract information from the mixed-format data (JSON in CSV) in fig. 1b

Fig. 4: RML field declarations for extracting information from the files in fig. 1.

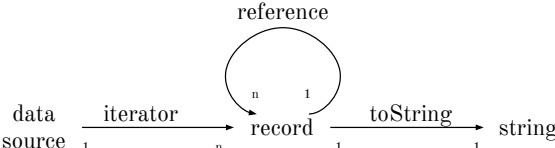

Fig. 5: Extracting information from data sources with RML fields. An iterator extracts $n$ records from a data source. One or more references extract $n$ further records from each record. A record has a string value.

To solve the mixed data format problem introduced earlier, we extend the reference formulation concept of RML to specify the formulation for each field separately. An example is shown on line 11 of fig. 4b: the `items` column can be referenced as CSV column, but the JSON objects stored inside can be processed using JSONPath. This process is similar to xR2RML's mixed-syntax paths.

### 3.2   Algorithmic representation

The information extracted from the input data can be represented in tables. We only introduce this representation as a tool for defining the semantics of RML fields, RML engines are not required to instantiate these tables. We choose the tabular format because it is easy to understand.

Since each reference can in theory return multiple values, we give each field a separate table, so that the extracted tables do not violate the 2nd normal form [3]. A field's table has three columns and these columns contain records, as well as provenance of which previous records created a record. Concretely, for each field we have these columns (shown in fig. 6, right-to-left):

– A column containing the field's records, one per row, as returned by the field's reference. This column has the same name as the field[3].
– A column containing an index of every record produced by this reference. This index makes it possible for records from this field to be referred to by records from this field's subfields. The name of this column is the name of the field concatenated with `.#`.
– A column containing references to the index column of the field's parent field (or to the iterator's index column for fields without a parent field). These references are used to keep track of which row in the parent field's or iterator's table the given row is based on: which record from the parent field was used as input to create the record in this row.

There is also one additional table for the iterator. This table contains two columns: one with the iteration index (column name #) and one with the iteration record (column name `it`). In fig. 6 the extracted tables for the iterator and for the fields `item` and `item.name` (all defined in fig. 4a) are shown.

The tables as defined for each field and the iterator can be joined to create one denormalized table for the logical source, containing all columns of all the source's fields' tables. The RDF generation is defined from this denormalized table. The logical source's denormalized table is defined as the natural, full outer join [4] of the iterator's table and all the fields' tables. By this definition records from different hierarchical levels in the input data that "belong together", i.e., records along the same root-to-leaf path in the input data's tree structure, will end up in the same row in the denormalized table. Therefore, RDF can be generated that mixes data from different levels of the input data's hierarchy without losing information about which data belongs together.

---

[3]  For nested fields, we consider `<parentname>.<declaredname>` as their name, with `<parentname>` the name of the parent field and `<declaredname>` the object of the field's `rml:name` property. For example: `item.name`.

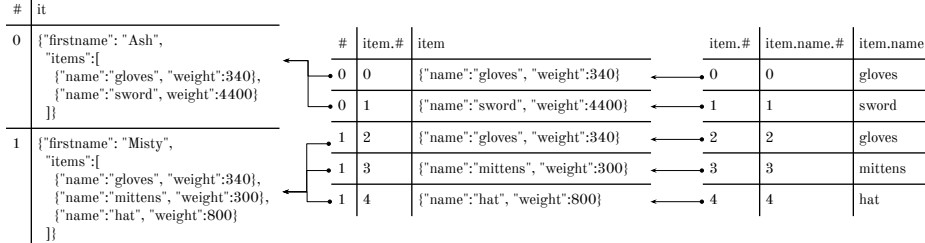

Fig. 6: The tree structure of the data in fig. 1a, here explicitly shown with arrows, is preserved by the index columns # and `item.#`.

This solution avoids concepts as "pushing down" values in a nested iteration (as in ShExML and xR2RML), but achieves the same goal of mixing input data from different hierarchical levels, while preserving the relation between related data. The join by which the denormalized table is defined should be an order-preserving join, as the order in the data source might need to be preserved in the generated RDF. The denormalized table for the running example is shown in table 2[4]. Again, this table is intended as a tool to specify the semantics if RML fields, an RML engine should not necessarily instantiate this table, especially as it might contain duplicates values in columns higher in the field hierarchy, as can for example be seen in the leftmost four columns in table 2.

| # | it | name.# | name | item.# | item | item.name.# | item.name | item.weight.# | item.weight |
|---|-----|--------|-------|--------|------|-------------|-----------|---------------|-------------|
| 0 | {...} | 0 | Ash | 0 | {...} | 0 | gloves | 0 | 340 |
| 0 | {...} | 0 | Ash | 1 | {...} | 1 | sword | 1 | 44400 |
| 1 | {...} | 1 | Misty | 2 | {...} | 2 | gloves | 1 | 340 |
| 1 | {...} | 1 | Misty | 3 | {...} | 3 | mittens | 1 | 300 |
| 1 | {...} | 1 | Misty | 4 | {...} | 4 | hat | 1 | 800 |

Table 2: Data along the same root-to-leaf path in fig. 1a ends up in the same row of this denormalized table, after being extracted by the fields in fig. 4a.

### 3.3    Writing to RDF

Writing information extracted from a source to RDF is done using RML triples maps. A triples map is a combination of term maps. Each term map is defined by a position (subject, predicate or object) and by a way to generate an RDF term from field values. A typical way a term map creates RDF terms from field values is the URI template: this is an URI with "gaps" which are filled in

---

[4]  The values in column `it` and `item` are omitted for brevity in table 2, but should be the same as those in fig. 6.

by field values. For example the URI template `:person/{name}` creates URIs from the value of the `name` field. The semantics of a triples map can be defined using the denormalized table from the previous paragraph: for each row in the denormalized table, fill in the term maps with the values of the corresponding columns and return the thus created triples. One caveat is that the denormalized table might contain NULL values introduced by the outer join. If a term map would be filled in with a NULL value, the triples created from this NULL value are omitted from the output.

In general, order and duplicates do not matter when generating RDF, since the RDF model itself is unordered and duplicate-free [6], so the duplicates introduced by the full outer join do not matter. However, some RDF constructs such as collections *are* affected by order and duplicates. Further, removing duplicates has been shown to positively affect performance of knowledge graph generation [12]. For these two reasons we introduce a duplicate-removal mechanism to select relevant duplicate-free segments of a logical source's denormalized table.

The duplicate removal works as follows: if a combination of subject, predicate and object map uses fields $f_1, \ldots , f_n$, then select from the denormalized table the distinct values in those columns ($f_1,...,f_n$) and their index columns ($f_1.\#,...,f_n.\#$), while preserving order. Triples for this subject, predicate and object map are then generated from the thus selected table, which contains all the required information, but no duplicates. An example of such a selected table is shown in table 3, for the three term maps `:person/{name} a :Person`, which together only contain a reference to one field: `name`. It should be clear that in many cases, the duplicate-free table needed for a triples map can be instantiated without first instantiating the denormalized table from which it is defined.

| name.# | name |
|--------|-------|
| 0      | Ash   |
| 1      | Misty |

Table 3: Duplicate-free table selected from table 2 for the field `name` and its index.

Finally, to create RDF collections we reuse the nested term map from xR2RML. A nested term map generates collections by grouping several generated terms into one list or set or other RDF collection. By default, terms are grouped into a collection based on the fields used in the other terms of the triples map, but other groupings could be explicitly declared by the user. For example, in fig. 7 we see that the items generated by the nested term map in the object position are grouped on the `name` field since that field is used by the subject term map.

## 3.4 Backwards-compatibility

As long as no nested fields are used, the fields extension of RML is exactly as expressive as "regular" RML. In fact, there is quite a simple equivalency

```
:triplesMap a rr:TriplesMap ;
  rr:subjectMap [
    rr:template ":person/{name}" ] ;
  rr:predicateObjectMap [
    rr:predicate :hasItems ;
    rr:objectMap [
      rr:termType rr:List ;
      xrr:nestedTermMap [
        rr:template ":person/{name}/item/{item.name}"
      ] ] ] .
```

(a) Nested term map used with RML fields to create lists of persons' items.

```
:person/Ash :hasItems
  ( :person/Ash/item/gloves
    :person/Ash/item/sword ) .
:person/Misty :hasItems
  ( :person/Misty/item/gloves
    :person/Misty/item/mittens
    :person/Misty/item/hat ) .
```

(b) Graph with RDF lists in the turtle syntax [2]. Created by the triples map in fig. 7a from the logical source in fig. 4a.

Fig. 7: Generating lists of `items` grouped per `person` with RML fields.

between regular RML on the one hand and RML with non-nested fields on the other. Regular RML references can be seen as "syntax sugar" [13] for RML fields in the following way. Given an RML mapping, each reference in a term map can be replaced by adding a field with the same reference and replacing the use of the reference by the name of the field. The obtained RML mapping *with* fields has exactly the same semantics as the original RML mapping.

```
:source1 a rml:LogicalSource ;
  rml:source :file ;
  rml:referenceFormulation ql:JSONPath ;
  rml:iterator "$.characters[*]" ;

:triplesMapPeople a rr:TriplesMap ;
  rr:subjectMap [ rr:template ":person/{$.name}" ] ;
  rr:predicateObjectMap [
    rr:predicate :hasName ;
    rr:objectMap [ rml:reference "$.firstname" ] ; ] ;
  rr:predicateObjectMap [
    rr:predicate :hasItem ;
    rr:objectMap [ rr:template ":person/{$.name}/item/{{$.items.[*]}.{$.name}}"]].

:triplesMapItems a rr:TriplesMap ;
  rr:subjectMap [ rr:template ":person/{$.name}/item/{{$.items.[*]}.{$.name}}" ] ;
  rr:predicateObjectMap [
    rr:predicate :hasName ;
    rr:objectMap [ rml:reference "{$.items.[*]}.{$.name}" ] ; ] ;
  rr:predicateObjectMap [
    rr:predicate :hasWeight ;
    rr:objectMap [ rml:reference "{$.items.[*]}.{$.weight}" ] ; ] .
```

Fig. 8: Syntax sugar for RML fields logical source in fig. 4a. References representing fields in fig. 4a are **bolded**. References representing *nested* fields are also *italicized*.

Similarly, we introduce a syntax for nested fields in the same style as regular RML. The idea is that this syntax is less verbose and more similar in style and spirit to regular RML than the syntax for nested fields from the start of this section. Of course this shorter syntax is again equivalent to using nested fields. In short, if an expression like `{ref1}.{ref2}` is used in a term map, it should be interpreted as if there is a field with reference `ref1` with a nested field inside with reference `ref2`. A more extended example of this and of the previous syntax sugar can be seen in fig. 8.

## 4 Validation

The W3C community group for Knowledge Graph Construction identified nine challenges for declarative mapping languages. Each challenge gives a high level problem statement of a feature current mapping languages cannot always successfully handle. There is also a set of input files and the expected output graph for each challenge. Of these nine challenges, five are related to nested iteration and references returning multiple values, namely: access-fields-outside-iteration, generate-multiple-values, multivalue-references, process-multivalue-reference and rdf-collections. We go over these five challenges, briefly describe each of them, and explain how RML fields allows to handle the challenge. We published our solutions to these challenges in a public github repository[5].

*Access fields outside iteration.* The challenge access-fields-outside-iteration relates to accessing data not directly present in the current iteration element. For example, sometimes during iteration over a lower hierarchical level, data in a higher hierarchical level is needed to mint an URI. RML fields solve this challenge by allowing to use field names from different hierarchical levels to create one term. The template `:person/{name}/item/{item.name}` from our running example (fig. 7a), creates URIs from person names (from level one of the input data hierarchy) and item names (from level two of the input data hierarchy). We could handle this challenge the same as our running example, since the input file for this challenge has a very similar structure to the file in our running example.

The challenge has an extension related to referring to data on the same iteration level but in different records, so-called "sibling" data. This challenge can be tackled by creating two fields with identical references, but different names: `id` and `friendId`. In different rows of the denormalized table, `id` and `friendId` will contain every combination of ids that occur in the data, since the content of these fields is defined by a full outer join. Saying a person is friends with all their siblings in the input data is then as simple as generating `ex:hasFriend` triples connecting people identified by `id` and `friendId`, as shown in fig. 9.

```
rml:field [
  rml:name "id" ;
  rml:reference "$.id" ] ;
rml:field [
  rml:name "friendId" ;
  rml:reference "$.id" ] ] ;
rr:subjectMap [
  rr:template "http://example.com/{id}" ] ;
rr:predicateObjectMap [
  rr:predicate ex:hasFriend ;
  rr:objectMap [
    rr:template "http://example.com/{friendId}" ] ] .
```

Fig. 9: By declaring a field with reference `$.id` twice with different names, all combinations of the reference's values can be linked by predicate `ex:hasFriend`.

---

[5] https://github.com/RMLio/mapping-challenges-rml-fields/

The challenge has a second extension about joining data in different data sets. While the RML fields extension as it is described in this document does not cover joins, we think the tabular algorithmic representation we introduced gives a solid foundation to add joins and other relational operators.

*Generate multiple values.* The generate-multiple-values challenge relates to generating multiple literals, each with its own language tag. This feature is typically needed when the input data stores strings together with the strings' languages.

RML fields handles this challenge by iterating over the multiple stored strings and their languages together. Then, using the standard RML feature of the language map, literals can be generated having the language tag stored in the input data, as shown in fig. 10.

```
rr:objectMap [
    rr:reference "firstname.label" ;
    rr:languageMap "firstname.lang" ]
```

Fig. 10: The label and language of `firstname` are iterated over together, so both can be used to create a single RDF literal.

This challenge has an extension related to default values for language tags. As it is more related to default behaviour of language maps and not to the iteration model, we did not focus on handling this extension when designing RML fields.

*Multivalue references.* The multivalue-references challenge relates to using references returning multiple values. This typically occurs when referring to tree-structured data. RML fields handles this challenge because it uses an iteration model where every reference can return multiple values.

The challenge has an extension related to generating triples that "skip" one or more iteration levels: for example, generating triples from data in levels one and three of the input hierarchy (skipping level two). This extension of the challenge is handled by RML fields since field names from any levels of the iteration can be used in triples maps. This includes using field names that skip a level in the same triples map, as shown in fig. 11.

```
rr:subjectMap [
    rr:template "http://example.com/lab/{name}" ] ;
rr:predicateObjectMap [
  rr:predicate ex:hasMember ;
  rr:objectMap [
    rr:template "http://example.com/author/{article.author.name}" ] .
```

Fig. 11: This triples map generates triples connecting data from the first level of the input hierarchy (`name`) to data in the third level (`article.author.name`), skipping the second level.

*Process multivalue references.* The process-multivalue-references challenge, like the previous one, relates to creating RDF from references returning multiple values, with an additional focus on extracting multiple values from strings and on generating different types of RDF collections.

RML fields handle extracting multiple values from strings by treating the strings as comma-separated value records and using the appropriate reference formulation for such records, as shown in fig. 12. Further, RML fields handle the generation of RDF collections by using xR2RML's nested term map, for which an RDF collection type can be specified.

```
rml:field [
  rml:name "author" ;
  rml:referenceFormulation ql:JSONPath ;
  rml:reference "$.authors.[*].name"
  rml:field [
    rml:name "firstname" ;
    rml:referenceFormulation ql:CSV ;
    rml:reference "1" ] ;
  rml:field [
    rml:name "lastname" ;
    rml:referenceFormulation ql:CSV ;
    rml:reference "0" ] ] .
```

Fig. 12: By changing reference formulation in nested fields, comma-separated values can be extracted from a JSON string.

*RDF collections.* The rdf-collections challenge focuses on the generation of RDF collections. As such, it overlaps with the previous challenge, but this challenge combines generating RDF collections with different structures for the input data.

Again, the generation of RDF collections is handled with RML fields by using xR2RML's nested term map. We find that using the nested term map allows to handle all cases in this more detailed challenge as well, one excerpt from a solution creating RDF lists is shown in fig. 13.

```
rr:objectMap [
  rr:termType xrr:RdfList ;
  xrr:nestedTermMap [
    rr:template "http://example.com/author/{article.author}" ] ;
] ;
```

Fig. 13: This object map creates RDF lists of authors.

## 5   Conclusion

We introduced the "fields" extension of RML that allows RML to generate knowledge graphs from nested data. We introduced this approach using a running example existing mapping languages cannot handle, but RML fields *can* handle. We validated our approach by showing it can now handle the

mapping challenges formulated by the W3C Knowledge Graph Construction Community Group related to nested data. Until now, no declarative mapping language could cover all the challenges we cover.

So far, we considered generating RDF from one data set with nested structure. However, in some use cases, information from multiple data sets needs to be combined during knowledge graph generation. This includes the "basic" case of joining two data sets on an attribute, but it also includes more advanced cases, such as generating RDF from data set A only if data set B contains no relevant information about an entity. This is exactly what we will investigate next. The herein introduced tabular algorithmic representation provides a strong basis for such combination of data sets, since many operators, such as join, leftjoin, etc., have been defined before for the tabular format.

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
