# OpenReview forum: "Integrating nested data into knowledge graphs with RML fields"
_eswc-conferences.org/ESWC/2021/Workshop/KGCW — KGCW 2021_

### Official Review · ~Herminio_García-González1 · 2021-04-01
**Good and interesting proposal with some weak points coming from the lack of implementation**

**Rating:** 7
**Confidence:** 5

**Review:**

This article presents a proposal called RML fields in the form of a new "syntax" for RML. Fields and nested fields are presented as a way to solve specific problems when dealing with hierarchical data and, therefore, solve 5 of the open challenges in the W3C Knowledge Graph Construction Community Group.

The strongest point of the paper and the proposal itself is how it builds into the existing solutions, takes benefit of them, integrates them, evolves and improves them into a proper RML mechanism which can effectively solve the existing problems. In addition, problems are well presented, it is easy to follow and it is well written. However, there are some minor issues that I would like to clarify and discuss with the authors.

In the introduction you accept that mixing contents can violate first normal form in databases. However, you also say that this kind of data is not unrealistic, and therefore, you aim to support them. Thus, is this something desirable or something that we should encourage (by supporting it)? (I think is quite justified in your paper but I want to establish a further discussion on that).

You do not capitalise cross-refs like Section, Table and Fig. Although it is not mandatory I would do it.

Listings along the paper are quite small, and because of that, when printed, bolded parts are not well distinguishable. You still have some space to make them a little bigger or use colors instead (e.g., red).

In Related work you mention: "However, it becomes difficult to refer to data stored in different hierarchical levels in the input data". I think that while this sentence is true is not quite informative to readers. I would suggest to explain JSONPath specification problems when accessing parent nodes. Right now it seems that is a problem of all hierarchical formats, and that is not true for XML and XPath.

In Fig. 2 and Fig. 3 you put two examples in xR2RML and ShExML but they are not offered as supplementary material so one can download them and test your claims. In addition for ShExML it would be nice to have a permalink to it using ShExML playground option for it. It would enhance reproducibility of your claims.

In Section 3.2 you propose an implementation to gather hierarchical information in a table, which solves the problem to access parent nodes. However, and in contrast with explicit solutions like in xR2RML and ShExML don't you think that it can be less performant? In other words, saving all this table information (just in case it is needed) could be critical in performance compared to counterpart explicit alternatives of saving only the needed parent nodes information.

Section 3.4: I have some concerns about this section. Firstly, I do not see that regular RML references can be seen as syntax sugar. We could define syntax sugar as some added syntax that makes some constructs easier to read or write but do not affect functionality. However, although it is not explicitly mentioned, I think that Fig. 8 cannot be executed nowadays looking to RML spec. Therefore, this is not backwards compatibility and, then, using current RML syntax, syntactic sugar would be totally the other way round. In addition, as I mentioned, I would have expected some discussion on real backwards compatibility using joins which would indeed be backwards compatible with existing engines. I am aware this is not possible with JSON files but it is with XML ones. I think [13] is not a good reference for syntax sugar.

In Fig. 9 I assume that the person is friend of him/herself :)

In the paragraph just below Fig 10. it is not clear whether it is supported or not nowadays.

In conclusions you say "Until now, no declarative mapping language could cover all the challenges we cover". As I understood from the paper you have a syntax/mechanism proposal but you don't have an implementation. Therefore, even though it is well explained, argued and grounded, you have no implementation to support that your theory works (I don't really think that for a workshop paper you must though). Although some other works only cover them partially they have working implementations which support their partial solutions. Therefore, I will be very cautious with triumphalist sentences like the mentioned one.

Some typos per section:

3.1 Fields

and each of these returned values is themselves -> and each of these returned values are themselves

3.2 Algorithmic representation

as a tool to specify the semantics if RML -> as a tool to specify the semantics of RML

5 Conclusion

using a running example existing mapping languages -> using a running example that existing mapping languages

We validated our approach by showing it can now -> We validated our approach by showing how it can now

---

### Official Review · ~Oscar_Corcho1 · 2021-04-02
**Interesting proposal for a syntactic RML extension (and some ideas on how to process it) to handle nested data**

**Rating:** 7
**Confidence:** 4

**Review:**

This paper presents a syntax extension to RML (which is conceptually called RML fields) that may be useful for dealing with some cases where data is not presented in a tabular format only, but in nested expressions (e.g., JSON objects), and hence the transformations into RDF require some changes to the traditional state-of-the-art transformations that are common in R2RML and RML.

I find the extension well motivated (I have had to deal in the past with similar cases, so I know well the limitations of existing works as well as the pre-processing steps that need to be done if one wants to use the existing approaches). It is also a sensible proposal, that builds on the experiences gathered in two previous proposals aiming at this direction, although not complete: xR2RML and ShExML. Table 1 is quite clear on all these relationships and how RML fields builds on them.

Now some comments that I hope that can be useful in order to improve the paper for this version or for future versions:
- In the end of section 2 you claim that your contribution is "a method to preserve the relation...". Honestly, I do not see it as a method, but as a syntactic approach to allow preserving all these cross-references without the need for the push-down and alike primitives used in the previous languages and approaches.
- I cannot understand the value of figure 5 in the context of a syntax + semantics description of a language, since it is operational, and not descriptive.
- Indeed, this is a common problem in section 3, and especially clear in section 3.3. When somebody presents a syntax and tries to provide the semantics for this extension of the language, there are several approaches to specify such semantics. I understand quite well the idea of transforming the problem into one of creating some (virtual, larger, but conceptually semantics-preserving) tables and then assuming that the semantics of RML will do the rest. Indeed, this would be a way to claim that the extension is no more than a syntactic sugar over the previous language, and not the opposite as it is claimed somewhere else. However, in section 3.3 you mix up the conceptual part of such mapping with the way in which it would be processed, talking for example about duplicate removal. This is not right, IMO, since one thing is how you are able to generate RDF from the pre-processed tables (even if they do not need to be generated by an RML-fields processor), and another thing would be talking about the computational and processing challenges that you would need to do. This part really needs some reworking if you want to keep separated the syntax from the semantics from the computational part.
- Indeed, associated to this previous comments, I would have expected some analysis of computational complexity or alike (not necessary for a workshop, IMO), before moving into how to process it (which is anyway not fully covered in this paper, again not needed for a workshop IMO).
- The validation section is nice, although I would have appreciated finding at least an alternative way in which people have dealt with such a challenge in the past (either with xR2RML or ShExML, or by some processing pipeline that generates intermediate representations of the data, as many people currently do). This would also provide you some support for further experiments in the future comparing your expected implemetnation with the existing approaches for this.

In summary, a good paper for this workshop, definitely worth being presented and discussed, but which would need some more cleaning and "reordering of ideas and how they are presented" in order to prepare it for a conference.

From a readability point of view, I would suggest increasing the fonts of the figures and examples, since they are difficult to read on paper.

Typo: "if RML fields" --> "of RML fields"

---

### Official Review · ~Franck_Michel1 · 2021-04-09
**Good proposition which would deserve a bit more details and clarifications**

**Rating:** 7
**Confidence:** 5

**Review:**

The paper proposes a rich set of extensions of the RML mapping language, grouped under the term "RML fields", designed to cover various mapping use cases: nested documents, mixed formats, and the generation of RDF collections.

The paper in well written and structured. The extensions are motivated by examples and validated against a set of use cases defined by the W3C KGC community group, thus attesting of the interest and validity of the extensions.

Overall this is an interesting contribution that will expand the applicability of RML to a wider scope of use cases.
I felt a bit confused while reading some of the contributions, and I think the paper could be a bit improved in this respect. I provide details below.

I basically have one concern about the way the backwards compatibility with regular RML is treated, and the implication it has with respect to the syntaxes that are allowed for references. I suggest to motivate and describe this much stronger, so as to avoid any confusion.


## About the backwards compatibility

Section 4.3 describes the backwards compatibility between RML fields and regular RML with extended references "{$.items.[*]}.{$.name}" (or "{{$.items.[*]}.{$.name}}" if this is in the context of a template). That makes sense.
But then, this raises a question: why would we need the rml:field extension at all if we are equally served with this extended reference notation?

I understand that the rml:field is required at least to support the mixed formats since one can "locally" change the rml:referenceForumlation, which is fine. But it kind of looks like you want to reconcile regular RML and RML with fields in a way that induces quite some confusion.

Furthermore, this introduces an ambiguity in the syntax that can be used as a reference: in one case rml:reference would take a JSONPath expression like "$.name"; but in the case where I define fields in the logical source, then rml:reference would take a field name that is simply "name".


## About the rml:field use

In fig 4a, you repeat all the fields that already exist, like [ rml:name "weight": rml:reference "$.weight" ]. Is it just an example or do you actually need to do it? If this is a requirement then this is quite cumbersome; instead, you could assume that the fields that are not mentioned explicitly will just be passed implicitly.

Also you describe the generalization of the reference in RML as a way to return records instead of values. This is ok. What about templates? Why not generalizing to them too? For instance, you could add fields by doing some sort of pre-processing like this:
rml:field [
    rml:name "fullName";
    rr:template "{*.firstName} {$.lastName}"
]


## Other remarks

The "Generate multiple values" challenge touches upon the variable language tag feature of RML. That seems out of scope since the question of multiple values is already covered by the "multivalues reference" challenge. I would recommend to remove this part.

In section 2, you write about xR2RML that "as the nested term map by definition generates individual terms, it does not account for cases where data from different hierarchical levels is used to generate more than one term in a triple".
Actually I'm not sure I get your point. There is no limitation in the number of terms that a nested term map can produce. Like term maps in xR2RML, a nested term map can return any number of terms.

Besides, you can specify an xrr:pushDown in the logical source together with the iterator.
Nevertheless, indeed xR2RML will not be able to generate the triples in fig. 1c because nested term maps work within predicate-object maps, but cannot be used to generate subject terms.


## Typos, misc.:
- In section 2 ref. to fig. 4a should be 1a.
- Add line numbers to listing 4 sice you refer to numbers in the text.
- "For example the iterator $.characters[*] returns a list containing the objects in the characters JSON array in ﬁg. 4a." => "For example, in ﬁg. 4a., the iterator $.characters[*] returns a list containing the objects in the characters JSON array of in ﬁg. 1a."
- Fig 7a: rr:List does not exist, you probably mean either rdf:List or xrr:List
- "this table is intended as a tool to specify the semantics **if** RML fields" : of RML fields

---

### Official Review · ~Pano_Maria1 · 2021-04-10
**Interesting approach that misses details**

**Rating:** 7
**Confidence:** 5

**Review:**

This paper introduces a new approach to solving the problems of generating values from hierarchial sources, combining elements from different nesting levels. It introduces a way to naturally group nested elements from different levels and extract values, without having to move elements up or down the hierarchy of the source.
Additionally it introduces a way to handle data with values in different reference formulations.

The paper builds on concepts from existing approaches to handle these challenges. Overall, the paper is well structured and builds up from problem and existing approaches,  to explaining the proposed solution, to showing how this solves several challenges proposed in the W3C Knowledge Graph Construction community group. However, I do feel that section 3 can be restructured to introduce the idea of the approach more clearly before diving into the details.

The presented approach does leave me with several open questions. The main questions:
- What is the relationship between the reference formulation in the logical source and field references in term maps?
- When using fields, is the original source iteration still available to use in that TriplesMap?
  - If so, how to differentiate between field references and normal reference formulation references?
- How does the syntax sugar for fields in term maps relate to the algorithmic representation introduced in 3.2?

If I have understood the approach correctly, to me it also raises some architectural concerns which should be considered:
- seperation of concerns between logical source iteration and term map evaluation
- not being able to use the inherent strengths of reference formulations in (table reference style) field representations. E.g. use of recursive queries in nested documents.

Overall I believe the presented approach is a very interesting, but is missing significant details on how it impacts the RML language. I believe this is in part due to the variety of problems that this paper attempts to tackle.  I do believe the approach warrants further discussion, testing, and fleshing out, and is a good paper for this workshop.

While reading, I've collected some additional remarks per section.

## Section 3.1, 3.2, 3.3:
NOTE: I've grouped these sections, since I could only understand section 3.1 after reading 3.2 and 3.3, and repeating that a couple of times. It might make sense to first introduce the concept of the algorithm a bit more clearly in section 3.1, before delving into the details. After reading just 3.1 and 3.2, I was left very confused on the approach.

These sections introduce the concept of RML fields and the generalization of the reference concept to afford further nested iterating. They present a promising mechanism for solving the problems of composite values and different iteration levels. However, reading these sections left me with the following questions:

- Why only use references to define a field? A template might also be interesting as a value generating expression on a field, not?
- What is the resulting iteration document structure that results from this logical source (fig. 4a)?
  Does a field add to an existing, document? Or does using fields mean the original document is not available any more, but only the tabular structure introduced by the fields? Or some other option?

The relationship between the reference formulation specified in the logical source and this algorithmic representation is unclear. One of the strong points of RML is that it is query language independent. Next to that there is a seperation of concerns between iterating over a document to produce sub-documents, which can then be used for value extraction.
However, this approach seems to change that.
If I understand it correctly, the idea is to (conceptually) create a single denormalized table from all fields in a given iteration record, and use the columns of that denormalized table for RDF generation. Furthermore the column name for a nested field will follow the pattern `<parentname>.<declaredname>`. In section 3.3 it is explained that this column name can be referenced to generate RDF value's.
This leads me to the conclusion that fields will be their own type of (tabular) source and have their own, currently implicit, reference formulation. This also means that you now get references in a mix of "field reference formulation" and the originally specified reference formulation (e.g. JSONPath or XPath). However, it does not become clear how these could be combined, if at all. Furthermore, the chosen syntax for field references could also be valid syntax in one of those reference formulations. E.g. `item.name` is valid JSONPath. How should an RML engine diferentiate between these, if combinable?

## Section 3.4:
In this section, syntax sugar for the use of RML fields is introduced. Although arguably the syntax could have been chosen differently, since it uses the same syntax elements as normal R2RML templates, the idea is great and this seems like a sane way to solve the problem of referencing nested data scoped to a certain node in the data. The relationship with section 3.2, though, is rather unclear here. How does this map to the tabular algorithm described in 3.2?

## Section 4:

**access fields outside iteration**:
The authors claim their approach solves the `access fields outside iteration` problem, including the extension relating to sibling data. While I can see how it solves the main problem, I do not see how it solves extension to sibling data. The solution proposed would also generate `ex:1 ex:hasFriend ex:1 .`, since the proposal describes no way to distinguish siblings from a current node.

**process multivalue references**:

In the described solution, a CSV reference formulation is used in nested fields, to process nested comma separated values. However, the `rml:reference` used in the term map, seems to reference an index: `rml:reference "1"` . This reference by index is not described in this paper, as far as I can tell. And as far as I can tell it is also not decribed in the RML specification, which introduces this reference formulation.
That being said, I can see that one could define such functionality, and then this approach would work.

## Textual suggestions / Typo's
- Add line numbers to snippets, for easier referencing
- Typo page 8: the semantics if RML -> the semantics of RML
- Typo page 8: duplicates values -> duplicate values
- Typo page 8, 10: an URI -> a URI
- Typo page 12 (fig. 10):  `rr:languageMap`  -> `rml:languageMap`.

---

### Meta-Review · Program_Chairs · 2021-04-21

**Recommendation:** Accept
**Confidence:** 5

**Metareview:**

The four reviewers agreed on the quality and the necessity of the paper. Although I would suggest to the authors to look in detail at the comments provided and integrate or include them in the camera-ready version or in (future) associated resources (i.e., spec?). This paper is important for both workshop and community group where I would expect a lot of discussions (and implication from the first two authors ;-)) on the next steps of this proposal and if there is a plan to incorporate it in the following versions of RML.

Good work,
David

---

### Decision · Program_Chairs · 2021-04-23

Accept